# Horse behavior and facial movements in relation to food rewards

**Laize G. Carmo, Laís C. Werner, Pedro V. Michelotto, Jr, Ruan R. Daros** ⓘ *

Graduate Program in Animal Science, School of Medicine and Life Sciences, Pontifícia Universidade Católica do Paraná, Curitiba, Paraná, Brazil

* r.daros@pucpr.br

**Data Availability Statement:** The original data, R scripts, and output can be found at URL: (https://doi.org/10.6084/m9.figshare.21714233.v1) DOI: (10.6084/m9.figshare.21714233.v1).

## Abstract

Food rewards are believed to have a positive valence in horses. The aim of this study was to assess the effect of food rewards on horse behavior before entering a horse chute, and behavior and facial movements while restrained in it. Thirteen female adult horses were brought once daily to an animal handling facility for three weeks. In week 1, baseline period, no reinforcement was applied. In weeks 2 and 3, experimental phase, half of the horses received positive reinforcement treatment after entering and remaining in the chute; the remaining horses were considered as controls (no positive reinforcement applied). There was a cross-over between the groups during the experimental phase. Horses were individually brought to the restraining chute and videos recorded during 60-sec. The duration and number of entries into the area close to the gate leading to the chute were measured before restraining and body posture, neck position, and tail swinging were recorded in the chute. Facial movements were also recorded and scored using EquiFACS methodology. Multilevel linear and logistic models were built to assess behavioral changes from baseline to the treatment phase and between phases (control and positively reinforced). Horses did not change their body posture or tail swings across the different phases ($P > 0.1$) and were less likely to show lowered neck during the positively reinforced phase (OR: 0.05; CI95%: 0.00–0.56; $P = 0.05$) compared to baseline. The likelihood of a lowered neck did not differ between the positive reinforcement and control phases ($P = 0.11$). In the positively reinforced phase, horses seemed to be more attentive (ears forward) and active (less eye closures, more nose movements) than in the control phase. A three-day positive reinforcement phase did not elicit major changes in body behavior in the chute but affected the facial movements of group-housed mares.

## 1. Introduction

Research on affective states of animals has historically focused more on negative than on positive affects, but the explicit incorporation of positive affects in animals into the animal welfare concept [1] has led to significant development of this topic in recent decades [2, 3]. Animal behavioral responses studies can provide insights into how an animal feels under a given

**Funding:** The authors received no specific funding for this work.

**Competing interests:** The authors have declared that no competing interests exist.

circumstance [4] and have been successfully applied to study positive affective states in animals [5, 6] including in horses [7].

Positive reinforcement can promote motivated behavior to access rewards and can be used to train animals to voluntarily participate in activities that are necessary (like veterinary care) but not pleasurable to the animal [8, 9]. Equids trained using positive reinforcement show fewer behaviors associated with discomfort, escape, and frustration and respond more rapidly to the activity imposed, expressing more behaviors associated with positive affective states than those trained with negative reinforcement [10, 11].

The analysis of facial behaviors has contributed extensively to the field of affective states of animals study. In equids, facial movements of pain have been described, resulting in the development of the Horse Grimace Scale (HGS) [12]. More broadly, the Equine Facial Action Coding System (EquiFACS) [13] allows the evaluation of all possible facial movements of horses, which is useful for a range of affective situations. However, little research has been done to date and most studies have only applied EquiFACS in the context of negative affective states like workday load [14], transportation and social isolation [15].

This study aimed to assess the behavioral and facial movements of horses being provided positive reinforcement through food rewards to execute a routine procedure. Horses are expected to change their behavior before entering the chute (e.g., spending more time close to the gate that leads to the horse chute) while body postures in the chute are expected to be associated with behaviors of expectancy (less resting, neck above withers) during the positively reinforced phase compared with the non-reinforced phase. Only few studies have assessed facial movements of horses in positive emotional contexts. Therefore, we only predict that horses will show differences in facial movements between the positive and non-reinforced phases.

## 2. Materials and Methods

### 2.1 Animal handling

This study was approved by the Animal Use Ethics Committee of the Pontifícia Universidade Católica do Paraná (CEUA-PUCPR #01749) and was conducted at the Fazenda Experimental Gralha Azul of PUCPR, located in the municipality of Fazenda Rio Grande, Paraná, Brazil. Thirteen healthy mixed-breed non-pregnant mares, ranging from 4 to 22 years of age, were used. The mares were housed together in large grazing paddocks with free access to shade and water. Hay and concentrates were fed twice daily.

Data was collected during the last three weeks of February 2021, on the afternoons of Wednesday, Thursday and Friday of each week. Week 1 was considered the baseline and habituation phase, and weeks 2 and 3 were considered the experimental phases. During the second week half horses received a food reward after entering the restraining chute and no food reward was supplied to the rest; a cross-over between these groups was applied in the third week (Fig 1). All horses were used as controls. Details of each phase are provided in the following sections.

During data collection days, at approximately 2 pm, all horses were walked from the grazing paddock to the horse management facility (Fig 2; approximately 150 m away from the paddock), consisting of a holding area (approximately 145 m$^2$) and a restraining area fitted with two similar horse chutes. Both areas were concrete floored. The horses had ad libitum access to water and hay in their holding area. The restraining area was fully covered, whereas the holding area covered 55% of its area.

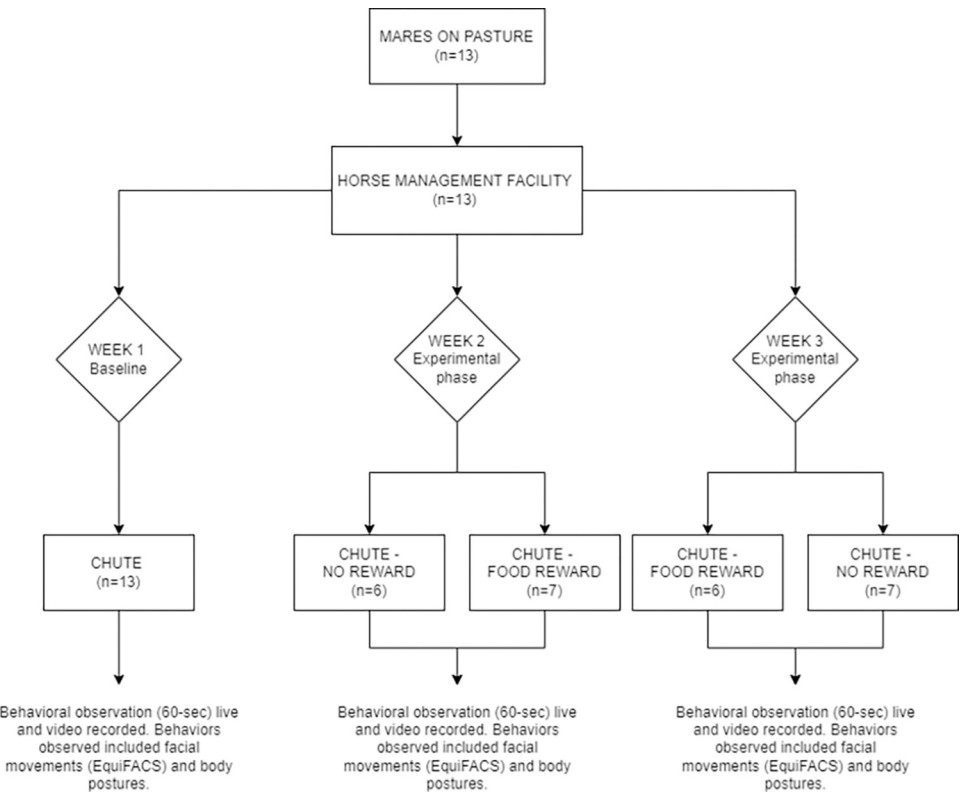

**Fig 1. Diagram of the study design.**

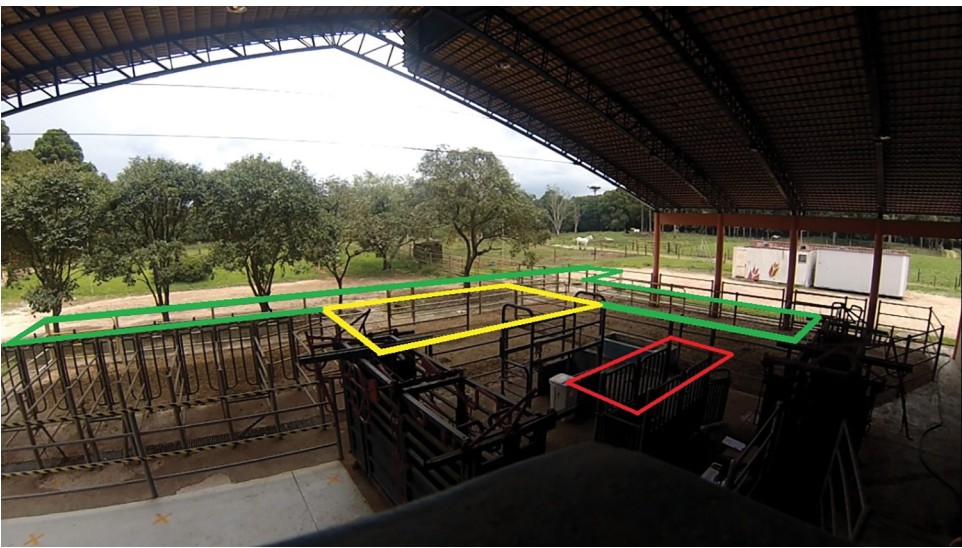

**Fig 2. Horse management facility.** In yellow is the central section of the holding area, surrounded by the peripheral section (in green). The restraining chute where the animals were handled for data collection is highlighted in red. Horses were allowed to roam freely between the sections of the holding.

## 2.2 Data collection setup

Three cameras were used for recording the study. A wide-lens camera (GoPro, model Hero3, GoPro Inc., California, USA) was positioned 4 m high adjacent to the management facility, to capture horse behavior in the holding area. The camera remained on during all data collection. To video record the behaviors in the horse chute, a hand camera (Sony, model DCR-PJ5, Sony Group Corporation, Tokyo, Japan) was placed at 1.1 m high, 1 m away from the back of the horse chute (Fig 3). Another camera (Canon, model T3i, Canon Inc., Tokyo, Japan) was positioned in front of the horse chute at a 45˚ angle, 1.5 m away from the chute and 1.5 m high, to capture the horse face (Fig 3). The video recording of these cameras (Sony and Canon) was synchronized and started when the horse entered the chute and stopped when the horse exited the chute.

The horses were free in the paddock before the start of data collection and were led in a group to the horse management facility. During the route, horses could assume different leadership positions and entry orders in the management facility according to their social configuration. The holding area of the management facility was divided into two sections. The section of the holding area that was closer to the gate accessing the restraining area was considered the central area, whereas the remaining sections of the holding area were considered peripheral (Fig 2). The horses were free to move between all sections and had access to hay and water from both peripheral and central areas. The GoPro captured the horses' movements in both sections.

After all the horses were in the management facility, they were taken individually by a lead rope to the chute. The chute entering order was pseudo-randomized (alternating between control and reinforced horses) and kept constant during the three weeks, i.e., horses were brought to the chute in the same order every day.

Before the study, horses were familiar to management facility but not to the feed rewards used in this study. Thus, one day before the beginning of the study, all horses were offered carrots, apples, and molasses treats. The procedure was performed twice. All horses promptly ate their carrots. Nine horses ate apples, and the molasses treat was eaten by three horses. None of the horses showed any sign of aversion (i.e., backing up after sniffing the food).

## 2.3 Habituation phase

This phase was completed during the first week of the study. The herd was brought to the management facility and allowed a period of 10 min without disturbance before the first horse was led to the restraining chute. All horses were familiar with the testing environment prior to the

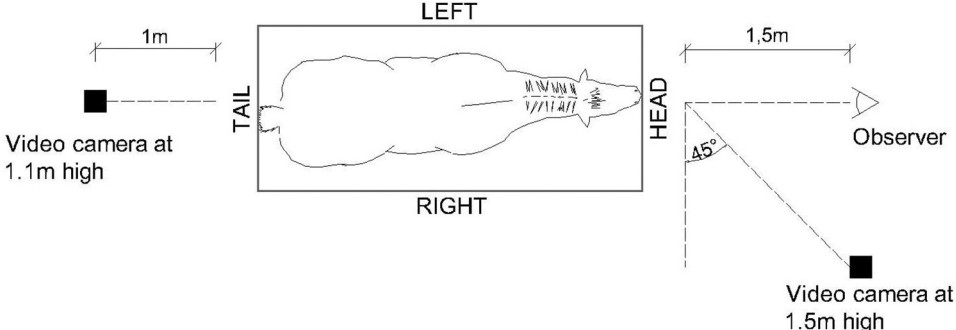

**Fig 3. Video camera and observer positioning for capturing body and facial behaviors of horses while in the restraining chute.**

experiment and were accustomed to entering the chute for veterinary treatment and daily reproductive management, but the individual experiences of each horse and their perceptions based on personality are unknown by the authors. A member of the research team familiar with the horses then entered the holding area, placed a halter on a horse, and led her to the chute. The horse was left in the chute without physical contact for 1 min. During this period, both cameras (Sony and Canon) were on, and the researcher observed the horse for selected behaviors live.

The observer stayed approximately 1.5 m away in front of the horse (Fig 3) without looking directly into her eyes and behaving in the most disinterested way possible. The videos were always recorded at the same place lasting 1 minute, and during video recordings, there was no movement of people or other animals close the horses. After the time in the chute, the horse was returned to the holding area and her halter removed. This procedure was repeated for each horse in this study.

## 2.4 Experimental phase

This phase was conducted in the second and third weeks of the study. The same handling procedures as described previously were used in this phase; however, seven horses received a food reward after entering the restraining chute and no food reward was supplied to the rest, i.e., they were in the control group in week 2. In week 3, a crossover was applied (i.e., the seven horses that received food rewards in week 2 were now in the control group and vice-versa).

The mares were offered a total of six food rewards (two small pieces of carrots, apples, and molasses treats). All food items were hand-fed one at a time starting with the carrot, apple, and then the molasses treat. If the horse refused any item, it was put aside and the next piece was offered. The rationale was to offer the horses a chance to eat or pass as they wish. The food items were offered immediately after the front gate of the restraining chute was closed and the horse was not pushing against the gates; thus, reinforcing the short sequence of behaviors that happens after the chute has been closed and the food offering. We highlight that the person closing the chute was not the same as the one offering the food item. This procedure and timing were applied consistently throughout the study.

Since chewing influences facial movements and other behaviors, the horses were recorded during the whole period that they were in the chute, but only a 60-sec video clip starting immediately after horses had finished chewing was used. The time spent in the chute from entering to release ranged from 4 to 6 min for horses in the experimental group. In the control group, no waiting period was applied, so the 60-sec video clip started immediately after the horse was closed in the chute. This approach was used to not create a situation that was novel and, possibly negative, to the horse (i.e., staying in the chute for 4 to 6 min without anything happening), potentially invalidating comparisons between phases. The observation time was standardized for each horse—i.e., a synchronized 60-sec video clip in both cameras (Sony and Canon).

## 2.5 Behavioral analysis

The holding area behaviors investigated in this study (evaluating data from the GoPro camera) were arrival order from the paddock to the holding area, time in each section of the holding area and frequency of entering the central section of the holding area before and after passing through the chute. Time variables were further categorized into a proportion, as horses who were first to enter the horse chute would, by default, have less time in the holding area before passing through the chute. Holding area behavioral data were assessed through video analysis by a single observer. Each video was analyzed for each horse and was manually recorded in a spreadsheet.

Table 1. Behaviors assessed during the restraining period in a horse chute.

| Behavior | Behavioral description |
|---|---|
| Standing | Standing with the weight resting on 4 legs. |
| Resting | Standing with the weight resting on 3 legs. |
| Pawing | Striking a vertical or horizontal surface or the air with forelimbs. |
| Kicking | One or both hind legs thrust backwards or to the side, contact with part of the stable may be achieved. |
| Rear | Front legs raised off the ground with forehand higher than hindquarters. |
| Neck above withers | Eye-level elevated above the height of withers, but below 45 degrees |
| Neck below withers | Eye-level parallel or below the height of withers. |
| Neck over 45 degrees | Neck raised over 45 degrees. |
| Tail neutral | Fleshy part of tail relaxed against the body. |
| Tail swish | Tail is flicked to one side and/or the other of the quarters. |
| Defecation/ Urination | Elimination of feces and urine. |

During the restraining period, several behaviors were recorded live and by video (Sony and Canon cameras). The selection of behaviors in the chute was based on and adapted from Young et al. [16]. Descriptions of each evaluated behavior are presented in Table 1. Behaviors described were collected continuously for 60-sec and annotated in a spreadsheet by an observer who was not blinded to the treatments and was always present being possible to observe the entire horse while in the chute. Pawing, kicking, rear and defecation/urination were recorded according to the frequency they occurred, while standing, resting and neck position and tail movement were recorded according to the duration.

Video clips from the last two days of the experimental phase (week 2 and 3, totaling 52 video clips of 60-sec) were analyzed to extract data regarding the facial action units (FAUs) of each horse in the chute.

A total of 17 Action Units (AUs) and 14 Action Descriptors (ADs) were assessed, following the EquiFACS methodology described by Wathan et al. [13]. The start and end of each assessed FAU was manually annotated by the same FACS coder watching on a computer. Each video was watched multiple times as only one FAU was observed and was rewound or slowed down when in doubt, being scored only when visible. Our objective was to evaluate the occurrence of all AUs during the 60-sec, however, in the moments when it was not possible to score AUs, no data was generated. We emphasize that most of the videos provided good visibility and good angle as the horses remained still. All the data were manually annotated using a spreadsheet. Facial recordings were only possible in the chute as the camera recording the holding area behavior was set to capture images from the whole herd.

The categorization of each FAU was done following the EquiFACS methodology [12], though we adapted the measurement of FAUs that are short lasting to measure frequency only (*eye closure*, *blink*, *half blink*, *lower lip depressor*, *chin raiser* and *lip presser*). FAUs that lasted longer were assessed as duration and frequency. The time required to correctly classify each FAU was always accounted for. In Table 2, we inform which FAUs were observed, using the codes AU, AD, AUH and EAD, as described in Wathan et al., 2015. The duration and frequency of each FAU were analyzed separately. Thus, the total duration (in seconds) and the frequency of each FAU are summarized per horse per day. The sequences of activation of each FAU and the co-occurrence during the 60-sec video clip were not considered.

Interobserver reliability was calculated (irr, R package) using data collected independently on five 60-sec video clips selected randomly from horses participating in the study. None of the

**Table 2. Facial action units (FAUs) assessed in the study using EquiFACS: The equine facial action coding system.**

| FAU[1] | Description |
|---|---|
| AU101 | Inner Brow Raiser |
| AU143 | Eye Closure |
| AU145 | Blink |
| AU47 | Half Blink |
| AU5 | Upper Lid Raiser |
| AD1 | Eye White Increase |
| AU10 | Upper Lip Raiser |
| AU12 | Lip Corner Puller |
| AU113 | Sharp Lip Puller |
| AUH13 | Nostril Lift |
| AU16 | Lower Lip Depressor |
| AD160 | Lower Lip Relax |
| AU17 | Chin Raiser |
| AU18 | Lip Pucker |
| AU122 | Upper Lip Curl |
| AU24 | Lip Presser |
| AU25 | Lips Part |
| AU26 | Jaw Drop |
| AU27 | Mouth Stretch |
| EAD101 | Ears Forward |
| EAD102 | Ear Adductor |
| EAD103 | Ear Flattener |
| EAD104 | Ear Rotator |
| AD29 | Jaw Thrust |
| AD30 | Jaw Sideways |
| AD133 | Blow |
| AD38 | Nostril Dilator |
| AD55 | Head Tilt Left |
| AD56 | Head Tilt Right |
| AD57 | Nose Forward |
| AD58 | Nose Back |

[1] Detailed description of each FAU can be found in Wathan et al. [13]. AU = Action Unit; AD = Action Descriptor; EAD = Ear Action Descriptor.

video clips were from the experimental phase of the study. In these videos, the frequency of each FAU was calculated and the intraclass coefficient (ICC) was used to determine the level of agreement between observers [17]. To account for the non-random selection of observers for this study and consistency across observations, the experimental design was such that the ICC was set for "two-way" and "consistency" as suggested by Hallgren [17]. ICC values ranged from -1 to 1 (complete disagreement to excellent agreement; zero being agreement no different than chance). The ICC was 0.73 (95% CI: 0.63–0.80); ranging from good to excellent agreement [17].

## 2.6 Statistical analysis

Statistical analysis was performed using R (version 4.0.4; R Core Team, 2021). Original data (. csv format) were imported into R and organized using the tidyverse package [18]. The original data, R scripts, and output can be found at https://doi.org/10.6084/m9.figshare.21714233.v1.

For numeric outcome variables and binary outcome variables (i.e., all behaviors except FAUs), mixed linear regression and mixed logistic regressions were applied, respectively [19]. As the study was divided into a baseline and an experimental phase, two sets of models were developed: 1) comparisons between the control and positive reinforcement phases, and 2) comparison between the baseline and the positively reinforced phase. Only the third day of the control and positively reinforced phases was used for comparison between data collected in the holding area, as that is the day we were most certain that the mare was habituated the experimental phase she was in. All three days of the baseline phase were used to compare the behaviors collected live and continuously. Horses were included as a random effect to allow for repeated measures. The order of treatments (control/reinforced) in the experimental phase was included in all models at first, but it did not affect the results ($P > 0.1$); thus, it was dropped from all final models. Age was also included but no effect was detected and thus dropped from the final models.

A similar statistical approach was used for each FAU; however, comparisons were only made between the control and positively reinforced phases using data from the last two days of each phase. The analysis of FAUs that presented high skewness (graphically assessed) was complemented with an additional generalized mixed model using Poisson distribution. The model assumptions of homogeneity and linearity were assessed graphically.

## 3. Results

### 3.1 Holding area behavior

It was expected that during the positively reinforced days, horses would arrive at the management facility before those that were under the control phase. However, this hypothesis was not confirmed because horses did not significantly change their arrival order (t = 0.44; $P = 0.66$) from the baseline to the positively reinforced days. When comparing the positive reinforcement with the control phase, no differences were detected in the order of arrival at the holding facility ($P = 0.80$).

There were no differences in the time spent in the central section of the holding area before passing through the chute between the control and positively reinforced phases (t = 0.87, $P = 0.39$). However, when comparing the last day of the positively reinforced phase against the baseline, there was a tendency for difference between these phases; during the positively reinforced days, horses spent 40% of the time in the central area before passing through the chute while they only spent 28% in the baseline phase (t = 1.81; $P = 0.08$).

On average, horses entered the central section of the holding area five times. This variable did not differ between the control and positively reinforced phases (t = 0.07; $P = 0.94$) or between the baseline and positively reinforced phases (t = 0.46; $P = 0.65$).

### 3.2 Behavior in the chute

When provided with a food reward, all horses ate at least part of the treats. They did not differ in most of the behaviors assessed while in the chute. However, defecation/urination, rearing, elevated neck, pawing, and kicking had very low frequencies or did not occur during any day of the study; therefore, they were not analyzed further (Table 3).

**Table 3. Total events of infrequently occurring body behaviors of all 13 mares observed during all sessions of 60-sec interval in a restraining chute.**

| Experimental phase / Behavior | Elevated neck | Pawing | Kicking | Rearing | Elimination |
|---|---|---|---|---|---|
| Positive reinforcement | 1 | 1 | 0 | 0 | 1 |
| Control | 4 | 2 | 1 | 0 | 0 |

Comparisons between the control and positive reinforcement phases and the baseline and positive reinforcement phase were performed to assess the frequency of lowered neck, resting all time, never resting, and tail swish. Only one behavior showed difference between the phases. Horses had 95% reduced odds of showing a lowered neck during the positively reinforced phase (OR: 0.05; CI95%: 0.00–0.56; P = 0.05) compared to the baseline phase. However, the likelihood of a lowered neck did not differ between the positive reinforcement and control phases (OR: 0.0; CI95%:0.00–38.0; t = -1.59; P = 0.11). We found no evidence of order effect of groups in any of our models.

### 3.3 Facial action unit—EquiFACS

The comparisons between the control and positively reinforced phases for each FAU are presented in Table 4. In brief, out of the 31 FAUs assessed (AUs and ADs), 20 presented differences between phases, seven did not differ between phases, and four were not observed in any phase. Among those that presented differences between the phases, there was a higher frequency or duration of *eye closure* (AU143), *inner brow raiser* (AU101), *lower lip relax* (AD160), *lips part* (AU25), *ear rotator* (EAD104), and *nostril dilator* (AD38) in the control phase. During the positive reinforcement phase, *eye white increase* (AD1), *lip corner puller* (AU12), *chin raiser* (AU17), *lip pucker* (AU18), *upper lip curl* (AU122), *jaw drop* (AU26), *ears forward* (EAD 101), *head tilt right* (AD55), and *nose forward* and *back* (AD57, AD58) were seen more often or for longer. *Mouth stretch* (AU27) was only observed in the positively reinforced phase. *Upper lip raiser* (AU10), *sharp lip puller* (AU113), and jaw thrust and sideways (AD29, AD30) were not observed during any phase.

Ear movements were the behaviors that lasted longest, with calculated mean during the positively reinforced phase showing that mares spent 50% of the observed time pointing their *ears forward* (EAD101), while during the control phase they spent only 23% of the observed time. Conversely, the *ear rotator* (EAD104) was activated for a longer period during the control phase (52%) of the time compared with the positively reinforced phase (25%). *Ear adductor* (EAD102) was observed on average for 20% of the time in both phases, and *ear flattener* (EAD103) was observed only once during the control phase and remained active for 20 s.

## 4. Discussion

Contrary to our hypothesis, horses did not spend more time in the central area of the handling facility during the reinforcement phase and there were no major differences in the behaviors displayed in the chute during the study, except for a reduction in the likelihood of horses showing a lowered neck in the positively reinforced phase. However, several specific changes in facial movements were detected in the positively reinforced phase.

### 4.1 Behavior in the holding area

It was expected that horses receiving food rewards would be more motivated to arrive faster at the management facility. The speed of relocation from the paddock to the facility could have been a better indicator of motivation, which could not be measured. However, there is no evidence that the horses changed their behavior as a function of food rewards. This may have been influenced by the low number of trials (three days) receiving food rewards or that the quantity or quality of the rewards was not sufficient to show marked behavioral changes. However, horses expecting a reward may approach a test area more rapidly than horses under standard management without being given rewards, which was shown within the first few days of the start of reward provisioning [20] and a study demonstrated that horses may also learn to

**Table 4. Estimated means for the positively reinforced (PR) and control phases from mixed linear regression models of each facial action unit (FAU) evaluated in 13 mares.**

| FAUs | Description | Duration (seconds) | | | Frequency | | |
|---|---|---|---|---|---|---|---|
| | | PR | Control | P-value | PR | Control | P-value |
| AU101 | Inner Brow Raiser | 6.5 ± 1.6 | 11.1 ± 1.6 | 0.05 [1] | 3.2 ± 1.5 | 3.8 ± 1.5 | 0.75 [1] |
| AU143 | Eye Closure | - | - | - | 0.2 ± 0.3 | 1.0 ± 0.3 | 0.03 [1] |
| AU145 | Blink | - | - | - | 18.7 ± 1.7 | 17.0 ±1.7 | 0.22 |
| AU47 | Half Blink | - | - | - | 6.4 ± 0.8 | 6.0 ±0.8 | 0.68 |
| AU5 | Upper Lid Raiser | 5.3 ± 1.9 | 3.5 ± 1.9 | 0.33 | 4.6 ± 1.1 | 1.6 ± 1.1 | 0.01 [1] |
| AD1 | Eye White Increase | 3.4 ± 1.1 | 2.0 ± 1.1 | 0.23 [1] | 1.9 ± 0.6 | 0.76 ± 0.6 | 0.06 [1] |
| AU10 | Upper Lip Raiser | NO[2] | NO[2] | - | NO[2] | NO[2] | - |
| AU12 | Lip Corner Puller | 2.1 ± 0.9 | 1.3 ± 0.9 | 0.35 [1] | 0.33 ± 0.2 | 0.16 ± 0.2 | 0.44 |
| AU113 | Sharp Lip Puller | NO[2] | NO[2] | - | NO[2] | NO[2] | - |
| AUH13 | Nostril Lift | 3.9 ± 2.9 | 7.8 ± 2.9 | 0.18 | 0.1 ± 0.1 | 0.2 ± 0.1 | 0.33 |
| AU16 | Lower Lip Depressor | - | - | - | 5.0 ±2.0 | 4.8 ±1.9 | 0.95 |
| AD160 | Lower Lip Relax | 24.9 ± 4.4 | 37.3 ± 4.3 | 0.03 | 4.0 ± 1.6 | 4.4 ± 1.6 | 0.81 |
| AU17 | Chin Raiser | - | - | - | 4.8 ±1.8 | 3.4 ±1.8 | 0.59 [1] |
| AU18 | Lip Pucker | 8.6 ± 2.5 | 2.8 ± 2.5 | 0.02 [1] | 4.2 ± 1.5 | 1.9 ± 1.5 | 0.17 [1] |
| AU122 | Upper Lip Curl | 3.3 ± 1.7 | 1.2 ± 1.7 | 0.40 [1] | 3.0 ± 1.6 | 1.0 ± 1.6 | 0.40 [1] |
| AU24 | Lip Presser | - | - | - | 8.6 ± 2.7 | 7.4 ± 2.7 | 0.75 |
| AU25 | Lips Part | 2.1 ± 2.8 | 9.2 ± 2.8 | 0.02 [1] | 1.2 ± 1.6 | 5.2 ± 1.6 | 0.05 [1] |
| AU26 | Jaw Drop | 5.7 ± 1.3 | 2.3 ± 1.3 | 0.06 [1] | 0.7 ± 0.9 | 1.34 ± 0.9 | 0.59 |
| AU27 | Mouth Stretch | 2.5 ± 1.3 | NO [2] | - | 2.7 ± 1.7 | NO [2] | - |
| EAD101 | Ears Forward | 30.2 ± 4.1 | 13.8 ± 4.0 | < 0.01 [1] | 3.2 ± 1.5 | 3.8 ± 1.5 | 0.79 |
| EAD102 | Ear Adductor | 12.0 ± 3.4 | 13.5 ± 3.3 | 0.65 [1] | 2.8 ± 0.7 | 2.0 ± 0.7 | 0.43 |
| EAD103 | Ear Flattener | NO [2] | 0.8 ± 0.6 | - | NO [2] | 0.8 ± 0.6 | - |
| EAD104 | Ear Rotator | 14.7 ± 4.0 | 31.2 ± 3.9 | < 0.01 [1] | 4.4 ± 1.4 | 4.6 ± 1.4 | 0.93 |
| AD29 | Jaw Thrust | NO [2] | NO [2] | - | NO [2] | NO [2] | - |
| AD30 | Jaw Sideways | NO [2] | NO [2] | - | NO [2] | NO [2] | - |
| AD133 | Blow | 0.38 ± 0.1 | 0.12 ± 0.1 | 0.14 | 0.2 ± 0.2 | 0.5 ± 0.2 | 0.32 |
| AD38 | Nostril Dilator | 9.4 ± 2.2 | 12.6 ± 2.2 | 0.29 [1] | 16.6 ± 2.2 | 16.4 ± 2.1 | 0.96 |
| AD55 | Head Tilt Left | 0.3 ± 0.1 | 0.1 ± 0.1 | 0.16 | 0.3 ± 0.1 | 0.1 ± 0.1 | 0.12 |
| AD56 | Head Tilt Right | 0.4 ± 0.1 | 0.1 ± 0.1 | 0.05 [1] | 0.6 ± 0.2 | 0.1 ± 0.2 | 0.04 [1] |
| AD57 | Nose Forward | 2.9 ± 0.8 | 0.5 ± 0.8 | 0.02 | 2.8 ± 0.7 | 0.6 ± 0.7 | < 0.01[1] |
| AD58 | Nose Back | 2.2 ± 0.5 | 1.2 ± 0.5 | 0.20 [1] | 1.3 ± 0.3 | 0.4 ± 0.3 | 0.04 [1] |

"—" Not assessed.

[1] This facial action unit presented a difference ($P < 0.05$) between phases when analyzed via mixed Poisson regression. All statistical details can be found in the Supplementary statistical files.

[2] Not observed.

discriminate between compartments with and without a reward within 1 d after being positively reinforced [21].

A variety of food types were used as rewards and allowed the horses to refuse any item if desired. Since all mares ate most of the food items offered in all trials, the food reward was adjudged as non-aversive. The social hierarchy may have influenced the location of the horses in the holding area; thus, horses motivated to approach the central section may have been discouraged from doing so by a more dominant mare [22, 23]. This effect may have been exacerbated by the small holding area. Social hierarchy or interactions were not measured in the holding area.

## 4.2 Behavior in the chute

Horses had a low probability of showing a lowered neck during the positive reinforcement phase. Neck position has been linked with excitation levels in horses; thus, elevated necks may reflect a state of high arousal, hypervigilance, or attention towards specific stimuli [24]. Considering that no sonorous stimulus was applied and there was no movement of people or any local noise, we ruled out that horses showed less lowered neck more often during the positive reinforced phase because of differences in attention-related behaviors between phases. Lowered necks have been associated with more relaxed and calm states [25, 26]. Thus, it is likely that horses being offered food rewards were more attentive and remained with their neck more elevated, but not hypervigilant.

## 4.3 Facial action units

To date facial expressions in horses have mostly been described in the context of painful or stressful experiences. However, certain features of the face have already been identified in a positive context [7, 27], although these studies have not did not use EquiFACS methodology. Lansade et al. [7] found that analyses of facial movements during grooming revealed significant differences between groups. However, a negative stimulus (spraying a mild perfume close to the horse) induced a more marked change in facial behaviors compared to positive (grooming) and neutral (observer close to the horse, without physical contact) contexts [27]. We emphasize that our study did not aim to evaluate the facial expression of horses, however, we presented the activation response (duration and frequency) of each FAU separately. Isolated facial movements can also be promising to identify a prominent action (either in duration or frequency) in situations of positive affective valence. We understand that only together could they reflect a "facial expression", therefore, in future studies, the FAUs should be analyzed in co-occurrence.

Ear position has been associated with emotional valence in horses [28]. During the reinforcement phase, horses spent more time with their *ears forward* (EAD101), suggesting that they were more attentive. This behavior has been linked to both positive [29] and negative [14] situations in horses. Thus, it is likely that forward ears alone are not sufficient to indicate the emotional valence of horses. Conversely, *backward ears* (EAD104) may reflect an aversive or painful state [11, 30, 31]. During the control phase, horses spent longer with their ears backward. To avoid any confounding with mares paying attention to other sources of noise or people movement, we carefully chose to conduct our data collection during the period of low area usage and guided all staff and students to stay clear during our data collection.

Horses have presented *nostril lifts* (AUH13) in slightly negative contexts, such as transportation and social isolation [14]. There were no differences in nostril lift between the study phases, but nostrils were more likely to be dilated for longer during the control phase, perhaps indicating that standing in a chute without stimuli away from the herd (although visual contact was possible) was negative for the horses. Conversely, the *nose forward* (AD57) and *nose back* (AD58) were observed more often in the positively reinforced phase. We are unaware of studies that have linked this behavior with an equine affective state.

Most of the studies that assessed eyelid movements did not provide a thorough description of the eyelid movement [32, 33], making it difficult to compare our results. We found that during the control phase, horses had higher frequencies of *eye closure* (AU143) compared to the reinforced phase (a movement different from blinking or half blinking). It is possible that the control horses closed their eyes more as a result of being in a more relaxed state or engaging in resting behavior. However, *blink* (AU145) and *half blink* (AU47) showed no difference between the phases. It seems that blinking frequency may not be a good indicator of emotional

valence as there are conflicting results regarding it. Increased blink frequency has been associated with slightly negative situations, such as transportation and social isolation [14, 32] but one study found that the total number of blinks was reduced in horses when social isolation or feeding was withheld for a short period [33]. Future studies should assess the role of arousal on blinking frequency.

*Lower lip relax* (AD160) was more active in the control than in the reinforced phase. In our study, videos were taken immediately after horses had swallowed the food rewards; therefore, the tension in the lips likely lasted for several seconds during the video recording phase. Indeed, when looking at the frequencies of lower lip relaxation between phases, horses showed no differences between phases.

*Head tilt right* (AD56) occurred more often and was longer in the positively reinforced phase. Although this may have been associated with laterality [34, 35], we are cautious of this interpretation, as in our study, the structures on both sides of the chute were different and may have attracted more horses immediately after receiving their food reward.

Our control phase consisted of a low-interaction phase, possibly putting horses in a neutral or low-arousal emotional state. It could be argued that horses had generally negative experiences in the chute before the study and experienced social isolation during the study, which may have masked the potential positive effect of the food reward. However, horses were not reluctant to enter the chute even in the baseline phase. Furthermore, the use of food rewards may not have contributed to horses' affective states in the way we expected. Future studies should assess horse preferences and motivations for accessing food rewards in comparison to other positive stimuli, such as grooming and verbal praise [36]. Blinded observers should be considered to reduce potential bias and the FAUs should be analyzed in co-occurrence. Finally, our positive reinforcement may only have affected the arousal level of the mares and did not specifically affect emotional valence.

## 5. Conclusions

The use of food rewards did not generate marked behavioral differences in horses and did not elicit major changes in behavior while restrained in a chute. However, several changes in horse facial movements were detected. After providing food rewards, horses entered a more attentive state with ears forward, less eye closure, and less nostril dilation but increased nose movement and more right head tilt, possibly indicating a feed-driven emotional state. However, we were unable to verify whether food reward affected the emotional valence of mares.

## Acknowledgments

We thank the Fazenda Experimental Gralha Azul (FEGA-PUCPR) staff for their help in setting up the experiment, and the students who helped with data collection: Alécia Mendes Araújo da Cunha, Alice Marega Reis, Ana Laura Gervinski, Bianca Barbosa, Camila Trojanovski das Neves, Jessica dos Santos Moreira, Lorena Tonetti Assis, Matheus Borges de Carvalho, and Renata Aline Gagliano. LGC received scientific scholarships (PUCPR-PAIC200590 and PUCPR-PAIC210069) from the Brazilian National Council for Scientific and Technological Development (CNPq) during the study period and manuscript writing.

## Author Contributions

**Conceptualization:** Laize G. Carmo, Ruan R. Daros.

**Data curation:** Laize G. Carmo, Laís C. Werner, Ruan R. Daros.

**Formal analysis:** Ruan R. Daros.

**Methodology:** Laize G. Carmo, Laís C. Werner, Pedro V. Michelotto, Jr, Ruan R. Daros.

**Project administration:** Ruan R. Daros.

**Resources:** Ruan R. Daros.

**Supervision:** Pedro V. Michelotto, Jr, Ruan R. Daros.

**Writing – original draft:** Laize G. Carmo, Ruan R. Daros.

**Writing – review & editing:** Laize G. Carmo, Laís C. Werner, Pedro V. Michelotto, Jr, Ruan R. Daros.

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
