## [Decision Letter · Decision Letter 0]

16 Feb 2023

PONE-D-22-34365Horse behavior and facial movements in relation to food rewardsPLOS ONE

Dear Dr. Daros,

Thank you for submitting your manuscript to PLOS ONE. After careful consideration, we feel that it has merit but does not fully meet PLOS ONE’s publication criteria as it currently stands. Therefore, we invite you to submit a revised version of the manuscript that addresses the points raised during the review process.

We look forward to receiving your revised manuscript.

Kind regards,

Chris Rogers

Academic Editor

PLOS ONE

Journal Requirements:

Additional Editor Comments:

Thank you for the submission. The reviewers have asked for some edits to improve the clarity on the experimental design. Please also see the comments regarding the use of language to describe outcomes.

Reviewers' comments:

Reviewer's Responses to Questions

**Comments to the Author**

1. Is the manuscript technically sound, and do the data support the conclusions?

Reviewer #1: Yes

Reviewer #2: Partly

2. Has the statistical analysis been performed appropriately and rigorously? 

Reviewer #1: Yes

Reviewer #2: I Don't Know

3. Have the authors made all data underlying the findings in their manuscript fully available?

Reviewer #1: Yes

Reviewer #2: Yes

4. Is the manuscript presented in an intelligible fashion and written in standard English?

Reviewer #1: Yes

Reviewer #2: No

5. Review Comments to the Author

Reviewer #1: This is a straightforward piece of working describing the behavioural and facial responses of horses being provided positive reinforcement through the provision of food rewards to a routine task. While few results were obtained, it nevertheless provides information in an area where little research has been done to date. The biggest comment I have to improve the paper would be to analyze the carryover effect as described below.

Introduction: well-written and succinct

Methods:

- L106 – was a lead rope used to lead the horses to the chute?

- L117 - Were the horses familiar with the testing environment prior to the experiment? Were they accustomed to entering the chute? If so, for what purpose – eg. veterinary treatment? The question is, what previous experience did they have with the chute and how might this affect their perception. This is alluded to on L344 but should be clearly defined in the methods.

- L135 – why did you offer apple and molasses treats when not all the horses preferred them (L112)?

- L146 – authors acknowledge the differences between the control and the treatment in regards to the holding time in the chute. Although perhaps not directly comparable, it seems a wise choice. How long were the video clips for the control while in the chute?

- L158 – authors comment the behaviours were collected continuously. Please indicate which behaviours were frequency counts, which were duration, and how long the duration was. Were all videos standardized to a certain length of time? This is better described for the FAUs and in table 2.

- L159 – the observer was not blind to the treatments – this should be addressed in the discussion

- L192 – did you standardize the length of time observed for each horse? As noted above, it is not clear how long the video clips were (I think they were all one minute, but this should be clearly stated). If the video clips were not standardized, this would greatly affect the results.

- L204 – did you account for any carryover effect for those horses who experienced the positive reinforcement phase the first week and then the control phase the second week? This may affect your comparisons between treatment and control. Also why did you use only the third day for comparisons of behaviours but used the second and third days for comparisons of the FAUs?

Results:

- L211-215 – does this first sentence not say the same thing as the second sentence?

- L231 – perhaps specify in the table caption that the behaviours listed in table 3 are ones that occurred infrequently

- L236 – the phrase “All but one showed differences between the phases.” is confusing – do you mean only one behaviour differed between phases? Because you only give results for lowered neck. Does this mean that none of the other behaviours differed between phases? Please clarify.

Discussion:

- L322 – perhaps amend to saying “slightly negative”

- L327 – can you speculate on why the control horses may have closed their eyes more? Was this indicative of resting behaviour (low arousal, not much going on)?

Reviewer #2: General Comments.

The purpose of this study is laudable and it is necessary that research focus on positive affects in animals, as also stated by the authors. This is however a difficult task, because the ground truth of affective states of animals is hard to achieve and that many different cognitive processes may influence the affective states.

This study aims at investigating the effect on behaviours of food rewards in horses (n=13). Horses were brought from the pasture to a test area every day for three weeks. From this area, the horses were brought individually to a chute, in a cross over design, where a food reward was given or not. Video was recorded in the test area and while in the chute after 3 days of intervention.

I am however not sure that I have understood the design correctly, new additions come here and there. Please clarify, maybe add a figure of the study design, and make sure that the abstract contains all necessary information about the study design, e.g. time in chute, the food rewarding schedule etc. Please also clarify the figure texts accordingly.

Please explain which behaviours you believe you reinforce by administering food in the chute? I am not an ethologist, but are you reinforcing something or just teaching the horses to get food? What is meant by “toggled”? Would the horses, that were habituated to get food, when in the chute, not get “frustrated” when they got no food? Can you then use the horses as their own controls, if one control group is neutral and the other is “frustrated” – negative valence affect?

Besides from the difficulty in understanding the study design, and therefore also how the data were produced, the use of “positive emotions” seems mixed up with concept of positive welfare. Explain why food should produce positive affects? Please refer to your reference 6 for terminology and revise the entire text according to published concepts.

6. Yeates JW, Main DC. Assessment of positive welfare: a review. Vet J. 2008; 175(3):

389 293–300. doi: 10.1016/j.tvjl.2007.05.009.

Please state hypotheses clearly. I can see that they pop up here and there, but please collect them, it greatly helps the understanding of your statistics.

Specific comments.

Line 15: Positive reinforcement can elicit positive emotions in animals. Please remove this statement from the abstract. At the best this statement remains controversial and should not be in an abstract. You could instead state that food rewards are believed to have a positive valence in horses.

Line 58 and onwards: Not true, please include e.g. the works by Lea Lansade et al., for example Facial expression and oxytocin as possible markers of positive emotions in horses. DOI 10.1038/s41598-018-32993-z

Line 120 and onwards: … The horse was left in the chute without physical contact for 1 min. During this period, both cameras (Sony and Canon) were on, and the researcher observed the horses for selected behaviours live. Was it possible to observe the entire horse while in the chute?

Line 156: During the restraining period, several behaviours were recorded live and by video. I am sorry if I didn’t get the design correctly, but was this not the case also for the baseline horses? And the single observer was not blinded? Why?

Line 162: Video clips from the last two days of the positive reinforcement and control phases (totalling 52 video clips) were analysed to extract data regarding the facial action units (FAUs) of each horse. How long were the clips for FACS annotation and how did you select them? I guess that not all of the video provided footage from a good angle, as the horse moved its head? Please therefore state which visibility codes you used or explain how you handled data when AUs were not possible to score. Again, regarding the design: you did only record for facial activity in the chute, not in the test area?

Line 165: The start and end of each assessed FAU was manually annotated by a researcher watching on a… Was it the same FACS coder for all videos? Or how many different coders were engaged? Which training did coders have – certification as FACS reader?

Line 170: Therefore, the frequency of each FAU was assessed but several FAU were also selected

(i.e., the ones that are expected to be activated for longer periods) to be measured as state events.

In FACS, there are no point events. Start and stop is recorded for each AU, indeed the observation of both the onset and offset if necessary to define a specific AU. For example: the eye blink, closure and half blink cannot be properly discriminated without determination of the duration. Further, your table contains a number of Action Descriptors, which are not considered AUs, they are composed of less defined muscle movements and may have another neurophysiological background than the 17 AUs. I think it may be valuable to have also the ADs in your study, but please describe what is what. Please revise table, text and statistics accordingly.

Line184: The ICC was 0.73 (95% CI: 185 0.63–0.80); ranging from good to excellent agreement. Please use the Wexler’s agreement, which is recommended for FACS analysis.

Line 302: see earlier comment for line 58.

Line 261: Table 4: Estimated means for the positively reinforced (PR) and control phases from mixed linear regression models of each facial action unit (FAU) evaluated in 13 mares. From the perspective of facial expressions, it is highly unlikely that one single action unit should be associated to a certain affective state. This is at least not the case for human emotions, or the affective components of pain or stress.

As mentioned earlier, the behaviour is the bottom line of all cognitive and other brain processes, learning, memory. It is unlikely that the 13 mares, being from 4 of age up to 22, would be cognitively equal. Therefore, I think it could be rewarding to look at individual horses and their responses to food. While some might be unaffected you might see characteristic changes in other, and this might be related to rank, age, horsonality etc. You have only 13 horses so I think this would be doable. You already did a huge job to collect the data

Line 304: Facial movements. You have chosen to discuss individual AU/ADs. This may not give real meaning as facial expressions, since you have no co-occurrence calculated. There is no single AU/AD where the frequency or duration can be associated as a marker for a certain affective state. At least this is not the case for humans. No facial expression consist of only one AU, which gives meaning if we believe that facial expressions are used for communication.

Line 327: ….horses had higher frequencies of closing the eyelid for more than half a second. I am confused: did you state earlier that you recorded eye blink as point events? Did you still measure duration? Please clarify further in the Material and Methods section.

I think the conclusion is sound, but wonder what you had expected – what would you have interpreted as indicative of positive valence, regarding facial expressions?

6. PLOS authors have the option to publish the peer review history of their article (what does this mean?). If published, this will include your full peer review and any attached files.

Reviewer #1: No

Reviewer #2: **Yes: **Pia Haubro Andersen

---

## [Author Response · Author response to Decision Letter 0]

5 Apr 2023

Dear editor and reviewers. Thank you for your time assessing our manuscript. The detailed responses to all comments can be found below or in the rebuttal letter attached in the submission system.

Authors' response to the comments:

Reviewer #1: This is a straightforward piece of working describing the behavioural and facial responses of horses being provided positive reinforcement through the provision of food rewards to a routine task. While few results were obtained, it nevertheless provides information in an area where little research has been done to date. The biggest comment I have to improve the paper would be to analyze the carryover effect as described below.

AU: Thank you for taking the time to review and to suggest improvements in our manuscript.

Introduction: well-written and succinct

AU: Thanks!

Methods:

- L106 – was a lead rope used to lead the horses to the chute?

AU: L106-107 “… they were taken individually by a lead rope to the chute”.

- L117 - Were the horses familiar with the testing environment prior to the experiment? Were they accustomed to entering the chute? If so, for what purpose – eg. veterinary treatment? The question is, what previous experience did they have with the chute and how might this affect their perception. This is alluded to on L344 but should be clearly defined in the methods.

AU: We now state the activities they were used to in the facility. L119-122 “All horses were familiar with the testing environment prior to the experiment and were accustomed to entering the chute for veterinary treatment and daily reproductive management, but the individual experiences of each horse and their perceptions based on personality are unknown by the authors.”. We left the information about their potential perception to the discussion as we were not able to assess it before the study.

- L135 – why did you offer apple and molasses treats when not all the horses preferred them (L112)?

AU: Because some animals may have their preferred food, some like them all and some may like it varied we opted for offering all food items to them. We added more to this section, see L139-141 “All food items were hand-fed one at a time starting with the carrot, apple, and then the molasses treat. If the horse refused any item, it was put aside and the next piece was offered. The rationale was to offer the horses a chance to eat or pass as they wish”.

- L146 – authors acknowledge the differences between the control and the treatment in regards to the holding time in the chute. Although perhaps not directly comparable, it seems a wise choice. How long were the video clips for the control while in the chute?

AU: Fixed. L150-151 “In the control group, no waiting period was applied, so the 60-sec video clip started immediately after the horse was closed in the chute”.

- L158 – authors comment the behaviours were collected continuously. Please indicate which behaviours were frequency counts, which were duration, and how long the duration was. Were all videos standardized to a certain length of time? This is better described for the FAUs and in table 2.

AU: Fixed. L167-171 “Behaviors described were collected continuously for 60-sec and annotated… …Pawing, kicking, rear and defecation/urination were recorded according to the frequency they occurred, while standing, resting and neck position and tail movement were recorded according to the duration”. 

- L159 – the observer was not blind to the treatments – this should be addressed in the discussion

AU: Added to the limitations of our study. L386 “Blinded observers should be considered to reduce potential bias”.

- L192 – did you standardize the length of time observed for each horse? As noted above, it is not clear how long the video clips were (I think they were all one minute, but this should be clearly stated). If the video clips were not standardized, this would greatly affect the results.

AU: Information added. L154-155 “The observation time was standardized for each horse - i.e., a synchronized 60-sec video clip in both cameras (Sony and Canon).”.

- L204 – did you account for any carryover effect for those horses who experienced the positive reinforcement phase the first week and then the control phase the second week? This may affect your comparisons between treatment and control. Also why did you use only the third day for comparisons of behaviours but used the second and third days for comparisons of the FAUs? 

AU: Thank you for this comment. We found no carryover effects as highlighted on L223-225 “The order of treatments (control/reinforced) in the experimental phase was included in all models at first, but it did not affect the results (P > 0.1)”. Regarding the use of the third day we now added the rationale for it in L219-221 Only the third day of the control and positively reinforced phases was used for comparison between data collected in the holding area, as that is the day we were most certain that the mare was habituated the experimental phase she was in.”

In previously analysis, we also used the second last day. Same results, so to keep it simple we decided to use only one day. We did not apply this same rationale for the EquiFACS analysis. Facial movements are much more subjective than the other behaviours measured, taken into consideration that the mean of samples is closer to the truth than the value of a single sample we have taken this rationale for this analysis, though we understand that this is more complex and may cause some confusion at first. Furthermore, the EquiFACS analysis we were looking at the immediate effect of having a food reward, so habituation to the experimental phase should influence less the results. 

Results:

- L211-215 – does this first sentence not say the same thing as the second sentence?

AU: Sorry for the misunderstanding. First sentence refers to the comparison between the reinforced and the baseline phase. L235-239 now reads as follows: “It was expected that during the positively reinforced days, horses would arrive at the management facility before those that were under the control phase. However, this hypothesis was not confirmed because horses did not significantly change their arrival order (t = 0.44; P = 0.66) from the baseline to the positively reinforced days. When comparing the positive reinforcement with the control phase, no differences were detected in the order of arrival at the holding facility (P = 0.80)”. 

- L231 – perhaps specify in the table caption that the behaviours listed in table 3 are ones that occurred infrequently

AU: Done. L256-257 “Table 3. Total events of infrequently occurring body behaviors of all 13 mares observed during all sessions of 60-sec interval in a restraining chute”.

- L236 – the phrase “All but one showed differences between the phases.” is confusing – do you mean only one behaviour differed between phases? Because you only give results for lowered neck. Does this mean that none of the other behaviours differed between phases? Please clarify.

AU: Sorry for the lack of clarity. L261 now states “Only one behavior showed difference between the phases”.

Discussion:

- L322 – perhaps amend to saying “slightly negative”

AU: Done. L352. 

- L327 – can you speculate on why the control horses may have closed their eyes more? Was this indicative of resting behaviour (low arousal, not much going on)?

AU: Agreed. Speculation added L360-363 “We found that during the control phase, horses had higher frequencies of eye closure (AU143) compared to the reinforced phase (a movement different from blinking or half blinking). It is possible that the control horses closed their eyes more as a result of being in a more relaxed state or engaging in resting behavior”.

Reviewer #2: General Comments.

AU: Thank you for your time in reviewing our manuscript. We tried our best in improving the clarity of our methods. Please find below our detailed responses.

The purpose of this study is laudable and it is necessary that research focus on positive affects in animals, as also stated by the authors. This is however a difficult task, because the ground truth of affective states of animals is hard to achieve and that many different cognitive processes may influence the affective states.

This study aims at investigating the effect on behaviours of food rewards in horses (n=13). Horses were brought from the pasture to a test area every day for three weeks. From this area, the horses were brought individually to a chute, in a cross over design, where a food reward was given or not. Video was recorded in the test area and while in the chute after 3 days of intervention.

I am however not sure that I have understood the design correctly, new additions come here and there. Please clarify, maybe add a figure of the study design, and make sure that the abstract contains all necessary information about the study design, e.g. time in chute, the food rewarding schedule etc. Please also clarify the figure texts accordingly.

AU: Details to the abstract have been added and a new figure added to the main manuscript. See L15, 19, 21, 22. and Figure 1.

Please explain which behaviours you believe you reinforce by administering food in the chute? I am not an ethologist, but are you reinforcing something or just teaching the horses to get food? 

AU: Added. L141-145 “The food items were offered immediately after the front gate of the restraining chute was closed and the horse was not pushing against the gates; thus, reinforcing the short sequence of behaviors that happens after the chute has been closed and the food offering. We highlight that the person closing the chute was not the same as the one offering the food item. This procedure and timing were applied consistently throughout the study.”.

What is meant by “toggled”? 

AU: Toggled was a mistake and this section has been rephrased.

Would the horses, that were habituated to get food, when in the chute, not get “frustrated” when they got no food? Can you then use the horses as their own controls, if one control group is neutral and the other is “frustrated” – negative valence affect?

AU: Thank you for this comment. When planning the experiment, we had to decide between having this issue of potential frustration of some animals or run an experiment that would be potentially confounded by time (i.e. all mares in the reinforced phase in the same week and control on another). We opted for the former as, at least, we could try to control for the effects of different emotions during the control phase using analytics method (i.e., including the order effect in our models). We highlight this now on L264: “We found no evidence of order effect of groups in any of our models”.

Besides from the difficulty in understanding the study design, and therefore also how the data were produced, the use of “positive emotions” seems mixed up with concept of positive welfare. Explain why food should produce positive affects? Please refer to your reference 6 for terminology and revise the entire text according to published concepts.

6. Yeates JW, Main DC. Assessment of positive welfare: a review. Vet J. 2008; 175(3):

389 293–300. doi: 10.1016/j.tvjl.2007.05.009.

AU: Thank you for the suggested literature. The manuscript was revised and the use of "positive emotions" was replaced by "positive affects" and statements that food produces positive affects have been removed.

Please state hypotheses clearly. I can see that they pop up here and there, but please collect them, it greatly helps the understanding of your statistics.

AU: Added. L56-62 “Horses are expected to change their behavior before entering the chute (e.g., spending more time close to the gate that leads to the horse chute) while body postures in the chute are expected to be associated with behaviors of expectancy (less resting, neck above withers) during the positively reinforced phase compared with the non-reinforced phase. Only few studies have assessed facial movements of horses in positive emotional contexts. Therefore, we only predict that horses will show differences in facial movements between the positive and non-reinforced phases.”.

Line 15: Positive reinforcement can elicit positive emotions in animals. Please remove this statement from the abstract. At the best this statement remains controversial and should not be in an abstract. You could instead state that food rewards are believed to have a positive valence in horses.

AU: Done. L15: “Food rewards are believed to have a positive valence in horses”

Line 58 and onwards: Not true, please include e.g. the works by Lea Lansade et al., for example Facial expression and oxytocin as possible markers of positive emotions in horses. DOI 10.1038/s41598-018-32993-z

AU: Reference has been added L38-41: “Animal behavioral responses studies can provide insights into how an animal feels under a given circumstance [4] and have been successfully applied to study positive affective states in animals [5, 6] including in horses [7].”.

Line 120 and onwards: … The horse was left in the chute without physical contact for 1 min. During this period, both cameras (Sony and Canon) were on, and the researcher observed the horses for selected behaviours live. Was it possible to observe the entire horse while in the chute?

AU: Detail added. L168 “… was always present being possible to observe the entire horse while in the chute”. 

Line 156: During the restraining period, several behaviours were recorded live and by video. I am sorry if I didn’t get the design correctly, but was this not the case also for the baseline horses? And the single observer was not blinded? Why?

AU: Sorry for the lack of clarity. In this paragraph we are specifically talking about the behaviour in the holding area, which is the pen were horse are used to wait before getting into the area were the restraining chute is (details in Figure 2). We now make this clear on L161-162 “Holding area behavioral data were assessed through video analysis by a single observer”. 

The only fully trained observer for this analysis was also the researcher responsible for conducting the Project, thus it was impossible to blind. We address this limitation in our discussion. L385-386: “Blinded observers should be considered to reduce potential bias and the FAUs should be analyzed in co-occurrence”.

Line 162: Video clips from the last two days of the positive reinforcement and control phases (totalling 52 video clips) were analysed to extract data regarding the facial action units (FAUs) of each horse. How long were the clips for FACS annotation and how did you select them? I guess that not all of the video provided footage from a good angle, as the horse moved its head? Please therefore state which visibility codes you used or explain how you handled data when AUs were not possible to score. Again, regarding the design: you did only record for facial activity in the chute, not in the test area?

AU: L179-183 “Each video was watched multiple times as only one FAU was observed and was rewound or slowed down when in doubt, being scored only when visible. Our objective was to evaluate the occurrence of all AUs during the 60-sec, however, in the moments when it was not possible to score AUs, no data was generated. We emphasize that most of the videos provided good visibility and good angle as the horses remained still”.

Yes. L184-185: “Facial recordings were only possible in the chute as the camera recording the holding area behavior was set to capture images from the whole herd”.

Line 165: The start and end of each assessed FAU was manually annotated by a researcher watching on a… Was it the same FACS coder for all videos? Or how many different coders were engaged? Which training did coders have – certification as FACS reader?

AU: All videos were watched by the same self-trained (using the EquiFACS material) observer. To assess if this observer was reliable we measured interobserver reliability against another independently self-trained observer. Specified in L178-179: “The start and end of each assessed FAU was manually annotated by the same FACS coder watching on a computer”.

Line 170: Therefore, the frequency of each FAU was assessed but several FAU were also selected

(i.e., the ones that are expected to be activated for longer periods) to be measured as state events.

In FACS, there are no point events. Start and stop is recorded for each AU, indeed the observation of both the onset and offset if necessary to define a specific AU. For example: the eye blink, closure and half blink cannot be properly discriminated without determination of the duration. Further, your table contains a number of Action Descriptors, which are not considered AUs, they are composed of less defined muscle movements and may have another neurophysiological background than the 17 AUs. I think it may be valuable to have also the ADs in your study, but please describe what is what. Please revise table, text and statistics accordingly.

AU: Sorry. We understand that in FACS there are no point events and that start and stop is recorded for each AU. Declaring AUs as one-off events has been removed. However, we also evaluated the activation frequency of each AU. We reviewed this section as described in L186-191: “The categorization of each FAU was done following the EquiFACS methodology [12], though we adapted the measurement of FAUs that are short lasting to measure frequency only (eye closure, blink, half blink, lower lip depressor, chin raiser and lip presser). FAUs that lasted longer were assessed as duration and frequency. The time required to correctly classify each FAU was always accounted for. In Table 2, we inform which FAUs were observed, specifying use the codes AU, AD, AUH and EAD, as described in Wathan et al., 2015. The duration and frequency of each FAU were analyzed separately”.

Line184: The ICC was 0.73 (95% CI: 185 0.63–0.80); ranging from good to excellent agreement. Please use the Wexler’s agreement, which is recommended for FACS analysis.AU: We considered using Wexler’s however ICC is a more robust statistical measure as it takes into account not only the ratio of agreement between observers but also the variance in the data, and account for likelihood of agreeing by chance. ICC is a well established method for interobserver reliability see Hallgren, 2012 (Tutor Quant Methods Psychol. 2012 ; 8(1): 23–34.). 

Line 302: see earlier comment for line 58.

AU: Context added to the discussion. L330-336 “To date facial expressions in horses have mostly been described in the context of painful or stressful experiences. However, certain features of the face have already been identified in a positive context [7, 27], although these studies have not did not use EquiFACS methodology. Lansade et al. (2018) found that analyses of facial movements during grooming revealed significant differences between groups. However, a negative stimulus (spraying a mild perfume close to the horse) induced a more marked change in facial behaviors compared to positive (grooming) and neutral (observer close to the horse, without physical contact) contexts [27]”.

Line 261: Table 4: Estimated means for the positively reinforced (PR) and control phases from mixed linear regression models of each facial action unit (FAU) evaluated in 13 mares. From the perspective of facial expressions, it is highly unlikely that one single action unit should be associated to a certain affective state. This is at least not the case for human emotions, or the affective components of pain or stress.

AU: Agreed. "Facial Expression" has been completely removed from the manuscript. We highlighted this in our discussion. See L336-341: “We emphasize that our study did not aim to evaluate the facial expression of horses, however, we presented the activation response (duration and frequency) of each FAU separately. Isolated facial movements can also be promising to identify a prominent action (either in duration or frequency) in situations of positive affective valence. We understand that only together could they reflect a “facial expression”, therefore, in future studies, the FAUs should be analyzed in co-occurrence”.

As mentioned earlier, the behaviour is the bottom line of all cognitive and other brain processes, learning, memory. It is unlikely that the 13 mares, being from 4 of age up to 22, would be cognitively equal. Therefore, I think it could be rewarding to look at individual horses and their responses to food. While some might be unaffected you might see characteristic changes in other, and this might be related to rank, age, horsonality etc. You have only 13 horses so I think this would be doable. You already did a huge job to collect the data

AU: We really appreciate this comment. In fact, we just finished a trial conducted by LGC (first author) where we have more tests to assess personality and their association with other behavioural processes (e.g., attention and memory bias). In earlier versions of our analysis we included age in our models and none were significant (We now mention this in L224-225). Unfortunately, we still do not have rank and personality data ready to re-run our analysis. For the facial movements we did not test this covariate as the data collection was planned to be conducted within individuals. 

Line 304: Facial movements. You have chosen to discuss individual AU/ADs. This may not give real meaning as facial expressions, since you have no co-occurrence calculated. There is no single AU/AD where the frequency or duration can be associated as a marker for a certain affective state. At least this is not the case for humans. No facial expression consist of only one AU, which gives meaning if we believe that facial expressions are used for communication.

AU: Agreed. That’s the reason we refrain to mention facial expressions. See earlier comment for line L336-341: “We emphasize that our study did not aim to evaluate the facial expression of horses, however, we presented the activation response (duration and frequency) of each FAU separately. Isolated facial movements can also be promising to identify a prominent action (either in duration or frequency) in situations of positive affective valence. We understand that only together could they reflect a “facial expression”, therefore, in future studies, the FAUs should be analyzed in co-occurrence”.

Line 327: ….horses had higher frequencies of closing the eyelid for more than half a second. I am confused: did you state earlier that you recorded eye blink as point events? Did you still measure duration? Please clarify further in the Material and Methods section.

AU: Sorry for the lack of clarity, we do not measure the duration. The rationale for this is in L186-189 and reads as follows: “The categorization of each FAU was done following the EquiFACS methodology [12], though we adapted the measurement of FAUs that are short lasting to measure frequency only (eye closure, blink, half blink, lower lip depressor, chin raiser and lip presser). FAUs that lasted longer were assessed as duration and frequency. The time required to correctly classify each FAU was always accounted for.”. We also rewrote part of the discussion that now reads (L359-361) now reads as follows: “We found that during the control phase, horses had higher frequencies of eye closure (AU143) compared to the reinforced phase (a movement different from blinking or half blinking)”. 

I think the conclusion is sound, but wonder what you had expected – what would you have interpreted as indicative of positive valence, regarding facial expressions?

AU: Unfortunately, we didn't have a clear hypothesis regarding facial movements. We address this in L60-62 “Therefore, we only predict that horses will show differences in facial movements between the positive and non-reinforced phases”. Although we found no clearly evident facial movements with food rewards, we can say that positive reinforcement affected the facial movements of group-housed mares.

---

## [Editor Report · Decision Letter 1]

9 May 2023

Horse behavior and facial movements in relation to food rewards

PONE-D-22-34365R1

Dear Dr. Daros,

We’re pleased to inform you that your manuscript has been judged scientifically suitable for publication and will be formally accepted for publication once it meets all outstanding technical requirements.

Kind regards,

Chris Rogers

Academic Editor

PLOS ONE

Additional Editor Comments (optional):

Thank you for the revised manuscript and attention to the reviewers comments.
---

## [Editor Report · Acceptance letter]

2 Jun 2023

PONE-D-22-34365R1 

Horse behavior and facial movements in relation to food rewards 

Dear Dr. Daros:

I'm pleased to inform you that your manuscript has been deemed suitable for publication in PLOS ONE. Congratulations! Your manuscript is now with our production department. 

Kind regards, 

on behalf of

Dr. Chris Rogers 

Academic Editor

PLOS ONE